

# Student evaluations of teaching: teaching quantitative courses can be hazardous to one's career

Bob Uttl[1] and Dylan Smibert[2]

[1] Department of Psychology, Mount Royal University, Calgary, Alberta, Canada
[2] Department of Psychology, Saint Mary's University, Halifax, Nova Scotia, Canada

## ABSTRACT

Anonymous student evaluations of teaching (SETs) are used by colleges and universities to measure teaching effectiveness and to make decisions about faculty hiring, firing, re-appointment, promotion, tenure, and merit pay. Although numerous studies have found that SETs correlate with various teaching effectiveness irrelevant factors (TEIFs) such as subject, class size, and grading standards, it has been argued that such correlations are small and do not undermine the validity of SETs as measures of professors' teaching effectiveness. However, previous research has generally used inappropriate parametric statistics and effect sizes to examine and to evaluate the significance of TEIFs on personnel decisions. Accordingly, we examined the influence of quantitative vs. non-quantitative courses on SET ratings and SET based personnel decisions using 14,872 publicly posted class evaluations where each evaluation represents a summary of SET ratings provided by individual students responding in each class. In total, 325,538 individual student evaluations from a US mid-size university contributed to theses class evaluations. The results demonstrate that class subject (math vs. English) is strongly associated with SET ratings, has a substantial impact on professors being labeled satisfactory vs. unsatisfactory and excellent vs. non-excellent, and the impact varies substantially depending on the criteria used to classify professors as satisfactory vs. unsatisfactory. Professors teaching quantitative courses are far more likely not to receive tenure, promotion, and/or merit pay when their performance is evaluated against common standards.

Corresponding author
Bob Uttl, uttlbob@gmail.com

## INTRODUCTION

Anonymous student evaluations of teaching (SETs) are used by colleges and universities to measure teaching effectiveness and to make decisions about faculty hiring, firing, re-appointment, promotion, tenure, and merit pay. Although SETs are relatively reliable when average ratings across five or more courses (depending on class size) are used, their validity has been questioned. Specifically, numerous studies have found that SETs correlate with various teaching effectiveness irrelevant factors (TEIFs) such as class size (*Benton, Cashin & Manhattan, 2012*), subject (*Benton, Cashin & Manhattan, 2012*), and professor hotness/sexiness (*Felton et al., 2008*; *Felton, Mitchell & Stinson, 2004*). However,

it is often argued that correlations between TEIFs and SETs are small and therefore do not undermine the validity of SETs (*Beran & Violato, 2005*; *Centra, 2009*). To illustrate, *Beran & Violato (2005)* examined correlations between several TEIFs and SETs using over 370,000 individual student ratings. Although they reported $d = 0.61$ between ratings of courses in natural vs social science, they further analyzed their data using regression analyses and concluded that course characteristics, including the discipline, were not important. They wrote: "From examining numerous student and course characteristics as possible correlates of student ratings, results from the present study suggest they are not important factors." (p. 599). Similarly, using Educational Testing Service data from 238,471 classes, *Centra (2009)* found that the natural sciences, mathematics, engineering, and computer science courses were rated about 0.30 standard deviation lower than courses in the humanities (English, history, languages) and concluded that "a third of a standard deviation does not have much practical significance". If so, one may argue, SETs are both reliable and valid and TEIFs can be ignored by administrators when making judgments about faculty's teaching effectiveness for personnel decisions.

However, SET research has been plagued by several unrecognized methodological shortcomings that render much of the previous research on reliability, validity and other aspects of SET invalid and uninterpretable. First, SET rating distributions are typically strongly negatively skewed due to severe ceiling effects; that is, due to a large proportion of students giving professors the highest possible ratings. In turn, it is inappropriate and invalid to describe and analyze these ceiling-limited ratings using parametric statistics that assume a normal distribution of data (i.e., means, $SD$s, $d$s, $r$s, $r^2$; see *Uttl (2005)*, for an extensive discussion of the problems associated with severe ceiling effects, including detection of ceiling effects and consequences of ceiling effects). Yet all of the studies we have examined to date do precisely that—use means, $SD$s, $d$s, $r$s, and $r^2$ to describe SETs; and to investigate associations between SETs and TEIFs.

Second, when making judgments about the practical significance of associations between TEIFs and SETs, researchers typically rely on various parametric effect size indexes such as $d$s, $r$s, and $r^2$ or proportion of variance explained and, after finding them to be small, conclude that TEIFs are ignorable and do not undermine the validity of SETs. However, it has been argued elsewhere that effect size indexes should be chosen based not only on the statistical properties of data but also on their relationship to practical or clinically significant outcomes (*Bond, Wiitala & Richard, 2003*; *Deeks, 2002*). Given that SETs are used to make primarily binary decisions about whether a professor's teaching effectiveness is "satisfactory" or "unsatisfactory", the most appropriate effect size indexes may be relative risk ratio (RR) or odds ratios (OR) of professors passing the "satisfactory" cut off as a function of, for example, teaching quantitative vs. non-quantitative courses rather than $d$s, $r$s, and $r^2$ (*Deeks, 2002*).

Third, researchers sometimes evaluate the importance of various factors based on correlation and regression analyses of SET ratings given by individual students (individual student SET ratings) rather than on the mean SET ratings given by all responding students in each class (class SET summary ratings). However, the proportion of variance explained by some characteristic in individual student SET ratings is not relevant to the effect

the characteristic may have on the class SET summary ratings that are used to make personnel decisions about faculty members. For example, *Beran & Violato (2005)* based their conclusion that various student and course characteristics "are not important factors" based on regression analyses over individual student SET ratings.

Accordingly, we re-examined the influence of one TEIF—teaching quantitative vs. non-quantitative courses—on SET ratings and SET-based personnel decision in a large sample of class summary evaluations from a midsize US university. We had two primary objectives. First, what is the relationship between course subject and SET ratings? Specifically, what is the distribution of SET ratings obtained by Math (and Stats) professors vs. professors in other fields such as English, History, and Psychology? Second, what are the consequences of course subject on making judgments about professors' teaching effectiveness? Specifically, what percentage of professors teaching Math vs. professors teaching other subjects pass the satisfactory cut-off determined by the mean SET ratings across all courses or other norm referenced cut-offs that ignore course subject?

In addition, we also examined how personnel decisions about professors might be affected if criterion referenced, label-based cut-offs were used instead of norm referenced cut offs. In many universities, SET questionnaires use Likert response scales where students indicate their degree of agreement with various statements purportedly measuring teaching effectiveness. Professors' teaching effectiveness is then evaluated against various norm-referenced cut offs such as the departmental mean, mean minus one standard deviation (e.g., 4.0 on 5-point scale), or perhaps a cut off determined by the 20th percentile of all ratings such as 3.5 on 5 point scale. In other universities, SETs use label based response scales where students indicate whether a particular aspect of instruction was, for example, "Poor", "Fair", "Good", "Very Good", and "Excellent". Here, if students rate professors as "Poor", then, arguably, to the extent to which SETs measure teaching effectiveness (a contentious issue on its own), a professor's teaching effectiveness is not satisfactory. If students rate a professor as "Fair", the plain meaning of this term is "sufficient but not ample" or "adequate" (*Merriam Webster Online, 2017*) or "satisfactory". Presumably, if students rate professors as "Good" or higher, professors should be more than "satisfactory" and those rated as "Excellent" are deserving of teaching awards. In contrast to Likert response scales, label-based response scales directly elicit clearly interpretable evaluation judgments from students themselves.

## METHOD

We obtained 14,872 class summary evaluations, with each representing a summary of SET ratings provided by individual students responding in each class in a US midsize university (New York University or NYU). In total, 325,538 individual student SET ratings contributed to the 14,872 class summary evaluations. The unit of analysis used in this study are the class summary evaluations. The class summary evaluations were posted on the university's website (www.nyu.edu), available to the general public (rather than to registered students only), and were downloaded in the first quarter of 2008. Table 1 shows the individual questions on the NYU SET forms used to evaluate teaching effectiveness on

**Table 1  SET questions, mean ratings, and standard deviations.**

| Question | M | SD |
|---|---|---|
| 1. How would you rate the instructor overall? | 4.37 | 0.55 |
| 2. How informative were the classes? | 4.26 | 0.52 |
| 3. How well organized were the classes? | 4.19 | 0.55 |
| 4. How fair was grading? | 4.14 | 0.55 |
| 5. How would you rate this course overall? | 4.09 | 0.57 |
| 6. How clear were the objectives of this course? | 4.12 | 0.52 |
| 7. How well were these objectives achieved? | 4.10 | 0.53 |
| 8. How interesting was the course? | 4.01 | 0.63 |
| 9. To what extent were your own expectations met? | 3.90 | 0.58 |
| Mean overall rating (across all items) | 4.13 | 0.50 |

**Notes.**
$N = 14{,}872$.

a 5-point scale where $1 = $ *Poor* and $5 = $ *Excellent*. The mean ratings across all nine items and course subject (e.g., English, Math, History) were extracted from the evaluations and used in all analyses. The SET evaluations included responses to other questions including questions on workload, labs, and course retake that are not considered in this report. No ethics review was required for this research because all data were available to general public in form of archival records.

## RESULTS

Table 1 shows the means and standard deviations for individual SET items across all 14,872 courses as well as the mean overall average (i.e., average calculated for each course across the 9 individual items). Item mean ratings ranged from 3.90 to 4.37 with $SD$s ranging from 0.52 to 0.63. The mean overall SET rating was 4.13 with $SD = 0.50$.

Figure 1 shows the smoothed density distributions of overall mean ratings for all courses and for courses in selected subjects—English, History, Psychology, and Math, including the means and standard deviations. This figure highlights: (1) distributions of ratings are negatively skewed for most of the selected subjects due to ceiling effects, (2) distributions of ratings differ substantially across disciplines, and (3) mean ratings vary substantially across disciplines and are shifted towards lower values by ratings in tails of the distributions. The density distributions in Fig. 1 were generated using R function density with smoothing kernel set to "gaussian" and the number of equally spaced points at which the density was estimated set to 512 (*R Core Team, 2015*).

Figure 2 shows the density distributions for Math (representing quantitative courses) and English (representing humanities, non-quantitative courses). The thick vertical line indicates one of the often used norm-referenced standard for effective teaching—the overall mean rating across all courses. The thinner vertical lines show the overall mean ratings for Math and English, respectively. This figure highlights that although 71% of English courses pass the overall mean as the standard only 21% of Math courses do so. The vast majority of Math courses (79%) earn their professors an "Unsatisfactory" label in this scenario.
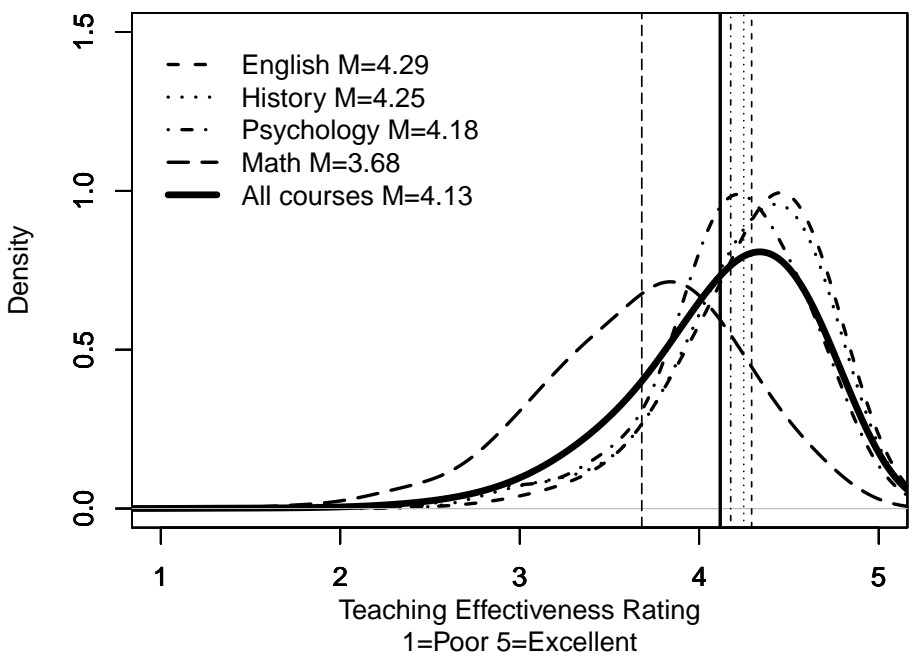

**Figure 1** **Distributions of overall mean ratings for all courses and for courses in selected subjects.** The figure shows the smoothed density distributions of overall mean ratings for all courses and for courses in English, History, Psychology, and Math, including the means and standard deviations. Figure highlights: (1) distributions of ratings are negatively skewed for most of the selected subjects due to ceiling effects, (2) distributions of ratings differ substantially across disciplines, and (3) mean ratings vary substantially across disciplines and are shifted towards lower values by ratings in tails of the distributions.

Figure 3 shows the same density distribution for Math and English but the vertical lines indicate criterion referenced cut-offs for different levels of teaching effectiveness—Poor, Fair, Good, Very Good, and Excellent—as determined by students themselves. It can be seen that Math vs. English courses are far less likely to pass the high (Very Good and Excellent) criteria.

Figure 4 shows the percentage of courses passing criteria as a function of teaching effectiveness criteria. If the teaching effectiveness criteria are set at 2.5 ("Good"), the vast majority of both Math and English courses pass this bar (96.60 vs. 99.63%, respectively). However, as the criteria are set higher and higher, the gap between Math and English passing rates widens and narrows only at the high criteria end where a few English and no Math courses pass the criteria.

Table 2 shows the percentages of course SETs passing and failing different commonly-used norm-referenced teaching effectiveness criteria as well as label-based criterion-referenced standards, for Math and English courses. The table includes the relative risk ratios of Math vs. English courses failing the standards. Math vs. English courses are far less likely to pass various standards except the label-based, criterion-referenced "Fair" and "Good" standards.

Finally, the mean overall SET rating for English courses was 4.29 ($SD = 0.42$) whereas it was only 3.68 ($SD = 0.56$) for Math courses, $t(828.62) = 22.10$, $p < .001$, $d = -1.29$
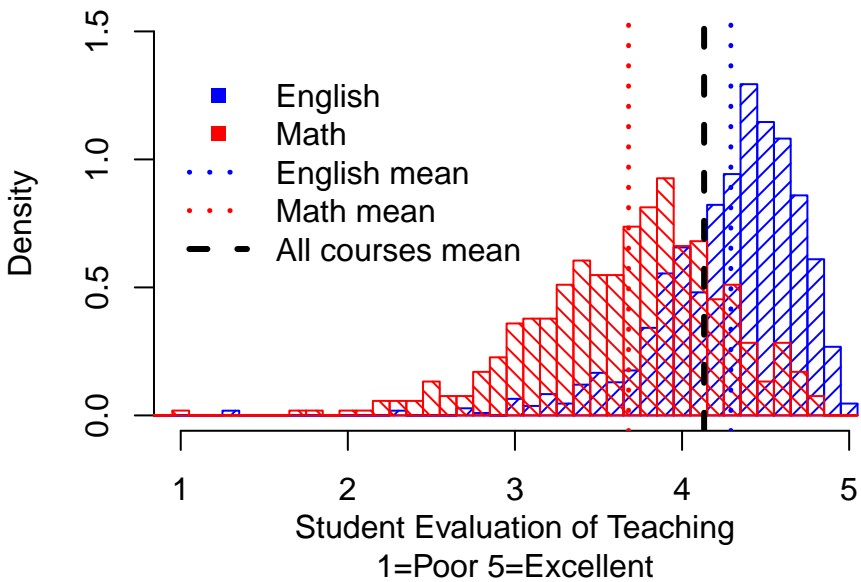

**Figure 2 Distributions of overall mean ratings for Math vs. English.** The thick vertical line indicates one of the often used norm-referenced standard for effective teaching—the overall mean rating across all courses. The thinner vertical lines show the overall mean ratings for Math and English, respectively. Although 71% of English courses pass the overall mean as the standard only 21% of Math courses do so. The vast majority of Math courses (79%) earn their professors an "Unsatisfactory" label in this scenario.

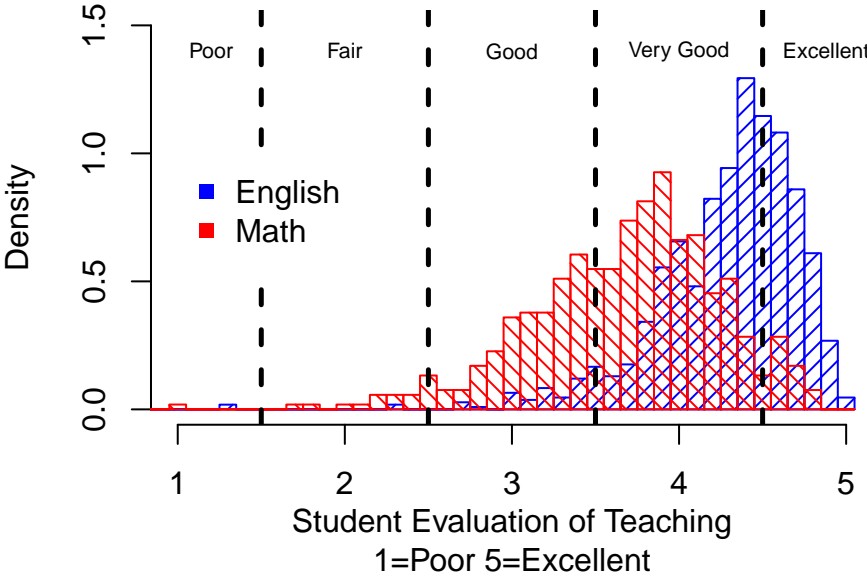

**Figure 3 Distribution of overall mean ratings for Math vs. English with criterion referenced cut offs.** The vertical lines indicate criterion referenced cut-offs for different levels of teaching effectiveness—Poor, Fair, Good, Very Good, and Excellent—as determined by students themselves. It can be seen that Math vs. English courses are far less likely to pass the high (Very Good and Excellent) criteria.

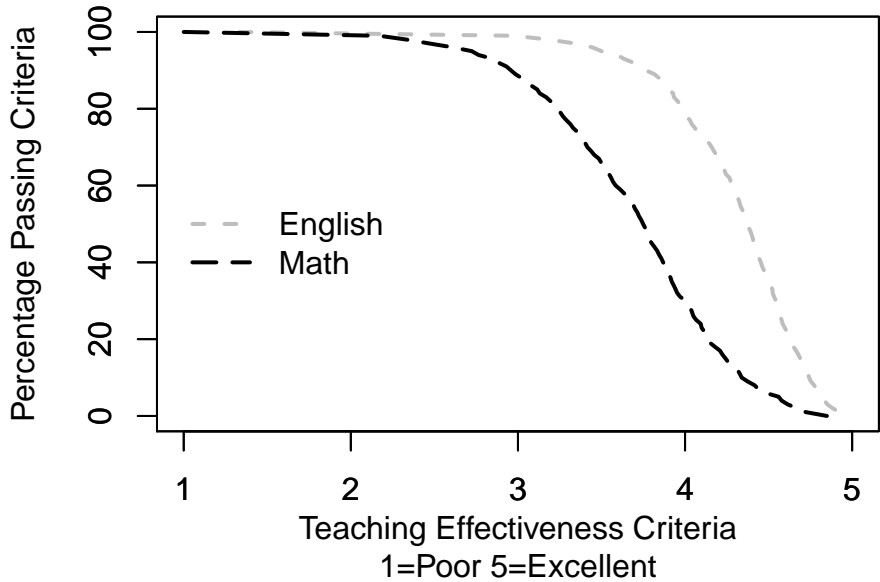

**Figure 4** **Percentage of courses passing criteria as a function of teaching effectiveness criteria.** If the teaching effectiveness criteria are set at 2.5 ("Good"), the vast majority of both Math and English courses pass this bar (96.60 vs. 99.63%, respectively). However, as the criteria are set higher and higher, the gap between Math and English passing rates widens and narrows only at the high criteria end where a few English and no Math courses pass the criteria.

**Table 2** **Percentages of course SETs passing vs. failing different SET standards and relative risk of failing to achieve the standards for Math courses.**

| Criteria cut-offs | All Pass (%) | All Fail (%) | Math Pass (%) | Math Fail (%) | English Pass (%) | English Fail (%) | *Math v. Non-Math RR of Failure incl. 95% CI* | *Math vs. English RR of Failure incl. 95% CI* |
|---|---|---|---|---|---|---|---|---|
| **Norm-referenced** | | | | | | | | |
| Mean (4.13) | 57.0 | 43.0 | 21.4 | 78.6 | 71.3 | 28.7 | 1.88[*] (1.80,1.98) | 2.74[*] (2.47,3.05) |
| Mean Minus 1 SD (3.63) | 84.5 | 15.5 | 58.0 | 42.0 | 93.1 | 6.9 | 2.89[*] (2.60,3.22) | 6.05[*] (4.76,7.70) |
| Mean Minus 2 SD (3.13) | 95.7 | 4.3 | 83.9 | 16.1 | 98.6 | 1.4 | 4.12[*] (3.34,5.09) | 11.59[*] (6.76,19.87) |
| **Criterion-referenced** | | | | | | | | |
| Excellent (4.50) | 25.4 | 74.6 | 5.9 | 94.1 | 35.4 | 64.6 | 1.27[*] (1.24,1.30) | 1.46[*] (1.39,1.53) |
| Very Good (3.50) | 88.5 | 11.5 | 66.0 | 34.0 | 94.9 | 5.1 | 3.17[*] (2.80,3.61) | 6.69[*] (5.04,8.89) |
| Good (2.50) | 99.3 | 0.7 | 96.6 | 3.4 | 99.6 | 0.4 | 6.02[*] (3.64,9.97 | 9.20[*] (3.13,27.06) |
| Fair (1.50) | 99.9 | 0.1 | 99.8 | 0.2 | 99.8 | 0.2 | 4.52 (0.54,37.47) | 1.02 (0.09,11.25) |

Notes.

All courses $N = 14,872$; English courses $n = 1,082$; Math courses $n = 529$; Non-Math courses $n = 14,343$.

[*]$p < .001$.

with 95% CI $[-1.18, -1.40]$. Critically, the correlation between the course subject (Math coded as 1, English coded as 0) and the overall mean rating was $r = -.519$, 95% CI $[-.553, -.482]$, $p < 0.001$, indicating that Math professors received lower ratings than English professors. In contrast, the correlations between the course subject (Math coded as 1, non-Math courses as 0) and the overall mean ratings when all non-math courses
are included, regardless of the degree of their quantitative nature, was relatively smaller, $r = -.172$, 95% CI $[-.188, -.156]$, $p < 0.001$, with $r^2 = 0.030$.

## DISCUSSION

Our results show that Math classes received much lower average class summary ratings than English, History, Psychology or even all other classes combined, replicating previous findings showing that quantitative vs. non-quantitative classes receive lower SET ratings (*Beran & Violato, 2005*; *Centra, 2009*). More importantly, the distributions of SET ratings for quantitative vs. non-quantitative courses are substantially different. Whereas the SET distributions for non-quantitative courses show a typical negative skew and high mean ratings, the SET distributions for quantitative courses are less skewed, nearly normal, and have substantially lower ratings. The passing rates for various common standards for "effective teaching" are substantially lower for professors teaching quantitative vs. non-quantitative courses. Professors teaching quantitative vs. non-quantitative courses are far more likely to fail norm-referenced cut-offs—1.88 times more likely to fail the Overall Mean standard, 2.89 times more likely to fail Overall Mean minus 1 SD standard—and far more likely to fail criterion-referenced standards—1.27 times more likely to fail the "Excellent" standard, 3.17 times more likely to fail the "Very Good" standard, and 6.02 times more likely to fail the "Good" standard. Clearly, professors who teach quantitative vs. non-quantitative classes are not only likely to receive lower SETs but they are also at a substantially higher risk of being labeled "unsatisfactory" in teaching, and thus, more likely to be fired, not re-appointed, not promoted, not tenured, and denied merit pay.

Regarding norm-referenced vs. criterion referenced standards, our results show that criterion-referenced standards label fewer professors as unsatisfactory than norm-referenced standards. Table 2 suggests that, in part due to substantially negatively skewed distributions of SET ratings, the norm-referenced cut-offs Overall Mean standard will result in 43.0% of classes failing to meet the standard, the Overall Mean minus 1 SD standard will result in 15.5% of classes not meeting it, and the Overall Mean minus 2 SD standard will result in 4.3% of classes failing this standard. In contrast, using students' judgments on the anchored scale, 99.3% of courses are considered "Good", "Very Good", or "Excellent" and only 0.7% of courses fail to meet "Good" standards in students' opinion. In other words, use of norm referenced standards results in labeling much greater percentages of professors as unsatisfactory than students themselves label as unsatisfactory. Moreover, professors teaching quantitative vs. non-quantitative courses are less likely to pass the standard under both types of standards.

Why has previous research often concluded that TEIFs, such as the courses one is assigned to teach, do not relate to SETs in any substantive way and were ignorable in evaluating professors for tenure, promotion, and merit pay? There are several methodological explanations: First, SET ratings often have non-normal, negatively skewed distributions due to severe ceiling effects. In turn, $d$s, $r$s and $r^2$ based effect size indexes are attenuated, invalid, and inappropriately suggest that influence of course subject on SETs is minimal. Second, parametric effect size indexes such as $d$s, $r$s, and $r^2$ assume normal distributions

and are inappropriate for binary "meets standard"/"does not meet standard" decision situations such as tenure, promotion, and merit pay decisions (*Deeks, 2002*). Third, some researchers used individual student SET ratings rather than class summary evaluations as the unit of analysis. However, using individual student SET ratings as the unit of analysis is inappropriate in this context because summative decisions are made based on class summary evaluations rather than on individual student evaluations.

In terms of inappropriate effect sizes such as $d$ or $r^2$, our results are generally larger than those reported by *Centra (2009)*, who used class summary evaluations from numerous institutions, and to those reported by *Beran & Violato (2005)*, who used individual SET ratings from a single university. We found $d = 1.29$ between Math vs. English SET ratings, *Centra (2009)* found $d = .30$, and *Beran & Violato (2005)* found $d = 0.60$ between "natural sciences" vs. "social sciences" SET ratings. Our correlational analysis showed $r = 0.18$ ($r^2 = 0.04$) between Math vs. Non-Math and SET ratings, whereas *Beran & Violato (2005)* found that this and other factors accounted together for less than 1% of the variance (i.e., $r^2 < 0.01$).

However, in contrast to previous research, we examined the impact of courses one is assigned to teach on the likelihood that one is going to pass the standard, and be promoted, tenured, and/or given merit pay and we found the impact to be substantial. Professors teaching quantitative courses are far less likely to be tenured, promoted, and/or given merit pay when their class summary ratings are evaluated against common standards, that is, when the field one is assigned to teach is disregarded. They are also far less likely to receive teaching awards based on their class summary SET ratings. The impact of using common standards may vary depending on whether a university uses the standards based on SET ratings of all professors across the entire university (university based standards) or the standards based on SET ratings of all professors within each department only (department based standards). If all or nearly all professors within the same department teach quantitative courses (e.g., math and statistics departments), the impact of using common vs. course-type specific department based standards to evaluate professors teaching quantitative vs. non-quantitative courses may be minimal. In contrast, if a few professors teach quantitative courses and the majority of professors teach non-quantitative courses within the same department (e.g., psychology, sociology), the impact of using common vs. course-type specific, department based standards may be as large or even larger than if university based standards were used.

Of course the finding that professors teaching quantitative vs. non-quantitative courses receive lower SET ratings is not evidence, by itself, that SETs are biased, that use of the common standards is inappropriate and discriminatory, and that more frequent denial of tenure, promotion, and/or merit pay to professors teaching quantitative vs. non-quantitative courses is in any way problematic. The lower SET ratings of professors teaching quantitative vs. non-quantitative courses may be due to real differences in teaching; that is, due to the professors teaching quantitative vs. non-quantitative courses being ineffective teachers.

However, lower SET ratings of professors teaching quantitative vs. non-quantitative courses may be due to a number of factors unrelated to professors' teaching effectiveness; for

example, students' lack of basic numeracy, students' lack of interest in taking quantitative vs. non-quantitative courses, students' math anxiety, and so on. Numerous research studies, task forces, and government-sponsored studies have documented steady declines in numeracy and mathematical knowledge of populations worldwide. For example, *Orpwood & Brown (2015)* cite the 2013 OECD survey showing that numeracy among Canadians declined over the last decade and that more than half of Canadians now score below the level required to fully participate in a modern society. We found that students' interest in taking quantitative courses such as introductory statistics was six standard deviations below their interest in taking non-quantitative courses (*Uttl, White & Morin, 2013*). Fewer than 10 out of 340 students indicated that they were "very interested" in taking any of the three statistics courses. In contrast, 159 out of 340 were "very interested" in taking the *Introduction to the Psychology of Abnormal Behavior* course. Moreover, this effect was stronger for women than for men: women's interest in taking quantitative courses relative to their interest in non-quantitative courses was even less than that of men. This lack of interest in quantitative courses propagates to lack of student interest in pursuing graduate studies in quantitative methods and lack of quantitative psychologists to fill all available positions. For example, the American Psychological Association noted that in the 1990s already there were on average 2.5 quantitative psychology positions advertised for every quantitative psychology PhD graduate (*APA, 2009*). If SETs are biased or even perceived as biased against professors teaching quantitative courses, we may soon find out that no one will be willing to teach quantitative courses if they are evaluated against the common standard set principally by professors who teach non-quantitative courses.

Thus, the critical question is: Are SETs valid measures of teaching effectiveness, and if so, are they equally valid when used with quantitative vs. non-quantitative courses or are they biased? Although SETs are widely used to evaluate faculty's teaching effectiveness, their validity has been highly controversial. The strongest evidence for the validity of SETs as a measure of professors' teaching effectiveness were so called multi-section studies showing small-to-moderate correlations between class summary SET ratings and class average achievement (*Uttl, White & Wong Gonzalez, 2016*). *Cohen (1981)* conducted the first meta-analysis of multi-section studies and reported that SETs correlate with student learning with $r = .43$ and concluded "The results of the meta-analysis provide strong support for the validity of student ratings as a measure of teaching effectiveness" (p. 281). *Cohen*'s *(1981)* findings were confirmed and extended by several subsequent meta-analyses (*Uttl, White & Wong Gonzalez, 2016*). However, our recent re-analyses of the previous meta-analyses of multi-section studies found that their findings were artifacts of small study bias and other methodological issues. Moreover, our up-to-date meta-analysis of 97 multi-section studies revealed no significant correlation between the class summary SET ratings and learning/achievement (*Uttl, White & Wong Gonzalez, 2016*). Thus, the strongest evidence of SET validity—multisection studies—turned out to be evidence of SETs having zero correlation with achievement/learning. Moreover, to our knowledge, no one has examined directly whether SETs are equally valid or biased measures of teaching effectiveness in quantitative vs. non-quantitative courses. Even the definition of effective teaching implicit in multi-section study designs—a professor whose students score highest

on the common exam administered in several sections of the same courses is the most effective teacher—has been agreed on only for lack of a better definition.

The basic principles of fairness require that the validity of a measure used to make high-stakes personnel decisions ought to be established before the measure is put into widespread use, and that the validity of the measure is established in all different contexts that the measure is to be used in *AERA, APA & NCME (2014)* and *APA (2004)*. Given the evidence of zero correlation between SETs and achievement in multi-section studies, SETs should not be used to evaluate faculty's teaching effectiveness. However, if SETs are to be used in high-stakes personnel decisions—even though students do not learn more from more highly rated professors and even though we do not know what SETs actually measure—fairness requires that we evaluate a professor teaching a particular subject against other professors teaching the same subject rather than against some common standard. Used this way, SET ratings can at least tell us where a professor stands within the distribution of other professors teaching the same subjects, regardless of what SETs actually measure.

## CONCLUSION

Our results demonstrate that course subject is strongly associated with SET ratings and has a substantial impact on professors being labeled satisfactory/unsatisfactory and excellent/non-excellent. Professors teaching quantitative courses are far more likely not to receive tenure, promotion, and/or merit pay when their performance is evaluated against common standards. Moreover, they are unlikely to receive teaching awards. To evaluate whether the effect of some TEIFs is ignorable or unimportant should be done using effect size measures that closely correspond to how SETs are used to make high-stakes personnel decisions such as passing rates and relative risks of failures rather than $d$s or $r$s. A professor assigned to teaching introductory statistics courses may find little solace in knowing that teaching quantitative vs non-quantitative courses explain at most 1% of variance in some regression analyses of SET ratings (*Beran & Violato, 2005*) or that in some experts' opinion $d = .30$ is ignorable (*Centra, 2009*) when his or her chances of passing the department's norm based cut off for "satisfactory" teaching may be less than half of his colleagues passing the norms.

## ACKNOWLEDGEMENTS

We thank Amy L. Siegenthaler and Alain Morin for careful reading and comments on the manuscript.

### Funding
The authors received no funding for this work.

### Competing Interests
The authors declare there are no competing interests.
## Author Contributions

- Bob Uttl conceived and designed the experiments, performed the experiments, analyzed the data, contributed reagents/materials/analysis tools, wrote the paper, prepared figures and/or tables, reviewed drafts of the paper.
- Dylan Smibert conceived and designed the experiments, performed the experiments, contributed reagents/materials/analysis tools, wrote the paper, prepared figures and/or tables, reviewed drafts of the paper.

## Data Availability

The raw data has been supplied as a Supplementary File.

## Supplemental Information

Supplemental information for this article can be found online at http://dx.doi.org/10.7717/peerj.3299#supplemental-information.

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
