# Peer review of "Student evaluations of teaching: teaching quantitative courses can be hazardous to one’s career"

_PeerJ, doi:10.7717/peerj.3299_

## Round 0.1 · original submission · Major Revisions

· Academic Editor

Major Revisions

Hi Bob:

I have received two thoughtful reviews of your article. Both reviewers, as well as I, believe that your article is interesting and newsworthy. However, both reviewers recommend revisions. I regard some of these revisions as things that must be revised and others as things that would be good to revise.

I will here briefly identify the must revisions, and leave it up to you to deal with the others.

In Point 1, Reviewer #1 asks how over hundred thousand students could have ended up providing less than 15 thousand evaluations. Clearly, this is confusing and needs clarification in the MS. What were the data you used for your analysis: individual student evaluations versus class-based summary stats?

In connection with Point 1, the reviewer writes that "this is an example of language in the paper that could be made more precise". I agree. When you have revised the paper, read it again a few more times and polish the language. This is especially true also in connection with Point 3 raised by the same reviewer.

In Point 2, Reviewer #1 asks for more details on how analyses were done. In view of the frequent discussion these days about replications, it is essential that manuscripts provide all of the details that would be required for replication. This pertains to the data you collected, the data sources, how they were summarized and analysed.

Point 4 goes to the limits of inferences that can be drawn from the data used in your study. The reviewer writes "This is a story, not science ...". I don't read this comment as saying we don't need the data you report. Rather, the reviewer is saying that the data permit no strong conclusions, but they can be used for speculating about a number of issues that come to the forefront when important decisions are made on the basis of students' ratings in different kinds of courses. Consistent with the reviewer's comment, I urge you to revise the manuscript so that its message does not come across as providing scientific proof that stats courses are harder or receive lower ratings, but rather as suggesting that there might be differences that deserve more careful scrutiny, reflection and interpretation.

Reviewer #2 makes comments that overlap with some of those from Reviewer 1. In addition, however, Reviewer #2 also recommends (Points 2 and 6) different ways in which the data might have been analyzed (multi-level modelling). I assume that you do not have access to the data that would be required for such alternative (more interesting) analyses. However, even if you do not have access to the data that would be required for such analyses, it would be useful/helpful to recommend a multi-level modelling approach as a way forward to address potential systematic difference in the ratings awarded in different college level courses.

Point 5 also deserves special attention and might be used to underscore the need for more cautious use of student teaching ratings when making career decision.

Let me end by pointing to Reviewer #2's first comment: "The sample size is fantastic ... [and] the topic is timely and important". We are aware that you may not have access to additional data that permit more sophisticated analyses, that you are stuck with the data that are available. All we ask is that you report these data more carefully, that you provide a more complete description of what you did with the data, and that you use your findings as fuel for a compelling story and speculations about the limits of student teaching evaluations, and the potential harm that their use might cause.

Reviewer 1 ·

Basic reporting

This manuscript makes several very important points:

1) SET play a crucial role in academic personnel matters.
2) SET vary by discipline and tend to be lower in quantitative courses, at least at one university.
3) "Explained variance" and fractions of standard deviations are not useful measures of differences across disciplines, in part because they do not take into account how SET are used.
4) In particular, if promotions, tenure, etc., are based on thresholds, small differences in mean scores can have a profound effect on careers.

That said, the article has serious statistical/scientific problems, among them:

1) How can 325,538 students produce only 14,872 course evaluations? Presumably, there are 14,872 _averages_, one per course, and the number of evaluations is far larger. This is an example of language in the paper that could be made more precise.

2) The authors do not describe the actual calculations they did. Simply saying, e.g., that the densities in figure 1 are "smoothed" does not make it possible to reproduce the work. They need to give far more technical detail. Reporting p-values without giving details of the tests performed is less than helpful.

3) The authors confuse "probability" with "frequency." There are no "likelihoods" or "probabilities" in their problem: there is a fixed census of data, no random sample, no experiment. The p-values are notional. All language concerning probabilities, likelihoods, etc., needs to be deleted and replaced with more circumspect language about frequencies in this particular data set. And generalizations to other universities or to SET in general needs to be more guarded and qualified.

4) The set up for the entire analysis only allows one (at best--see below) to draw the conclusion that, for this university, SET averages differ systematically between quantitative and non-quantitative courses. In particular, the design does not make it possible to conclude that the difference is unfair: if could be due to real differences in teaching, not to subject matter per se. (Personally, I have no faith in SET whatsoever as a measure of teaching quality--I think they perhaps measure satisfaction or enjoyment; perceived gender, attractiveness, and ethnicity of the instructor, and similar traits that should not mater pedagogically--but this study doesn't add any evidence either way.) The crucial question is whether "equally good" teaching is rated differently across disciplines. But there is simply no way to control for teaching effectiveness/quality, in part because it is nearly impossible to measure teaching effectiveness. This is a story, not science, and it gives no insight into the reason for the differences. Rather, it concludes that instructors of quantitative courses are disadvantaged by SET. Even though I believe the conclusion, I don't see that it can be justified on the basis of _this_ work.

5) There is no consideration of response rates, class sizes, class types (seminar v. lecture, required v. elective, ...), etc. It would be interesting (and perhaps could explain the results) if response rates were lower in quantitative courses than in non-quantitative courses or if quantitative courses tend to be larger and less personal. Cultural differences among students in different disciplines could potentially explain differences in response rates and in responses. Whether a course is required or an elective could be confounded with whether the course is quantitative. The level and size of the course could be confounded with whether the course is quantitative. And there could be systematically worse teaching in quantitative courses, although again I don't think that's what's going on.

6) The authors criticize the use of parametric statistical measures, but do not perform any nonparametric tests. Worse, they apparently report parametric p-values themselves, and for tests (e.g., correlation involving binary outcomes, apparently evaluated using standard Pearson tests; and differences in means, apparently evaluated using Student's t test) that are misleading or meaningless because the underlying assumptions are wrong in the current application. The notion of "effect size" for a Likert scale is troubling. The authors could instead perform nonparametric permutation and randomization tests to support, for instance, their claim that "distributions of ratings differ substantially" between quantitative and non-quantitative courses. The paper basically reports summary statistics, but treats them as if they are statistical tests.

7) Finally, the ms. needs to be proofread carefully.

Experimental design

See above.

Validity of the findings

See above.

Additional comments

The paper makes some excellent general points, but the quantitative claims need more serious statistical support, the analysis needs to be described in enough detail to be reproducible, and the language needs to be cleaned up to use words such as "likelihood" more carefully. There needs to be some acknowledgment of the limitations of the design (or lack of design), and a careful discussion of potential sources of confounding. And the parametric tests should be deleted--they are not appropriate.

Reviewer 2 ·

Basic reporting

No comments

Experimental design

There is much to like about this manuscript. The sample size is fantastic and, as someone who teaches quantitative courses, the topic is timely and important. Several notes on the present manuscript.

1. The effect sizes based on previous research found medium to large effect sizes (d = .61) based on Cohen’s interpretations. It is quite reasonable to castigate conclusions that these are not important factors.

2. The data presented are observational and instructors are not randomly assigned to topics or courses. Science courses may often have hard curves that are not present in humanities courses. There are many factors present here that may be related to teaching evaluations that cannot be accounted for. Some additional discussion of this would be helpful. Nonetheless, the point made by the authors is that teaching evaluations are used based on their results without any such adjustment (this point should be documented more strongly) and consequently the numbers themselves are what are used.

3. There are a number of ways of considering and presenting the data. The present approach is persuasive, but the authors should consider a slightly more sophisticated approach if feasible. I would suggest a multilevel model where rating is predicted by type of course with class and professor as random variables. This provides more useful information. To what extent do professors differ from each other? What is the difference in means between types of courses? Having the variability across professors (if professor is available as a variable in the data) would provide a nice counterpoint to understanding the difference in mean evaluations as a function of the type of course. This analysis may be more easily conducted separately for each type of class (e.g., English, History, Psych, Math) as the variances may not be constant across disciplines.

4. Figure 2 is hard to see. I would suggest box plots instead for English and Math with horizontal lines for the five categories. Box plots for Figure 1 would be easier to interpret and understand as well.

5. The weakest part of this manuscript is the criterion referenced cut-offs. In many contexts “Good” may not be actually good. Language for promotion and tenure often use words such as superior and excellent. This has no bearing on the argument made in the manuscript that there are real and substantial differences between math and english course ratings.

6. The manuscript would be strengthened if courses within different disciplines (e.g., Psychology, Sociology, etc) were classified as quantitative vs non quantitative. As well, many courses in Math are lower level requirements (e.g., calculus). What happens when one considers only upper level courses that are restricted in general to majors? Are these differences a function of lack of desire and aptitude on the part of the students?

Validity of the findings

no comments.

Additional comments

no comments.

---

## Round 0.2 · Minor Revisions

· Academic Editor

Minor Revisions

Dear Bob:

The revised version of your manuscript has now been assessed by the two people who also commented on the earlier version. Both reviewers like the revised manuscript, which now acknowledges the limited conclusions that can be drawn from the data set you worked with.

Both reviewers highlight the need for additional careful proof reading. As it stands, the manuscript still has a substantial number of grammatical and punctuation errors. Reviewer 1 identified a number of such errors. However, this list is not exhaustive; such errors occur throughout the manuscript.

In addition, Reviewer 2 recommends that you augment the manuscript by including confidence intervals for effect sizes, and he/she suggests ways to accomplish this goal.

The additional revisions should be easy to accomplish, and when done, I will be able to make a final decision on the manuscript.

I look forward to the final version of the manuscript.

Peter

Reviewer 1 ·

Basic reporting

I appreciate the additions the authors made to the manuscript. I recommend accepting it, but it still needs a careful reading for grammar and punctuation. Please see below.

Also, many universities compare SET means only within departments, not across departments. It would be acknowledge that on pp6-7, where the issue is discussed. There, it might also be worth mentioning gender bias and other biases that still would not be alleviated by limiting comparisons to within-department averages.

Typos/fragments/etc.:

Sentence fragment on p.5:
Clearly, professors who teach quantitative vs. non-quantitative classes are not only likely to receive lower SETs but they are also in substantially higher risk of being labeled “unsatisfactory” in teaching, fired, not promoted, and

p5:
(Centra, 2009) who used class summary evaluations from numerous institutition as well as those reported
-->
(Centra, 2009), who used class summary evaluations from numerous institutions, and to those reported

p6:
statistics was six standard deviation below -->
statistics was six standard deviations below

A fewer than 10 out of 340 students -->
Fewer than 10 out of 340 students

Moreover, this effect was stronger for women vs. men. Women were even more disinterested in taking quantitative courses than men relative to non-quantitative courses.
-->
Moreover, this effect was stronger for women than for men: women's interest in taking quantitative courses relative to their interest in non-quantitative courses was even less than that of men.

(p. 281) . --> (p. 281). (extra space)

Experimental design

observational study, not an experiment

Validity of the findings

Now that there are some caveats, this is fine.

Additional comments

It really needs careful editing.

Reviewer 2 ·

Basic reporting

Basic reporting is excellent.

Experimental design

The basic design is observational and the sample size is outstanding.

Validity of the findings

The primary research questions are not amenable to experimental design and the limitations of the present research are well documented.

Additional comments

There are several minor improvements that could be made. First is to provide confidence intervals for the effect sizes. These could easily be produced from resampling given that the authors have the raw data. The present data provide a medium to large effect size given Cohen's norms (d=.50 is medium and d=.80 is large). These are not small effects (difference between math and english SETs). The present effects (and Centra's 2009 effects) are comparable to classic effect sizes in social and personality research. See Funder and Ozer (1983). Finally, for observational data the raw correlation is a measured of shared variance. See Ozer (1985, 2007). These are minor changes.

Funder, D.C., & Ozer, D.J. (1983). Behavior as a function of the situation. Journal of Personality and Social Psychology, 44, 107-112.
Ozer, D. J. (1985). Correlation and the coefficient of determination. Psychological Bulletin, 97(2), 307-315.
Ozer, D. J. (2007). Evaluating effect size in personality research. Guilford Press.

---

## Author Rebuttal · Round 0.2

# Rebuttal Letter

Dear Peter:

Thank you for yours and the reviewers' feedback. We have now revised the manuscript following the feedback whenever possible. Our detailed responses are below.

Cheers,
Bob

## REVIEWER 1

**Re: Point 1 -- a question as to "How over hundred thousand students could have ended up providing less than 15 thousand evaluations"**

We thought we were clear that our unit of analyses were course evaluations rather than evaluation provided by individual students. However, clearly, we were not. We have now revised the manuscript in several places to highlight that the unit of analysis were class based course evaluation rather than individual student course evaluations.

**Re: Point 2 -- Not enough detail is provided for calculations (e.g., smoothing).**

The revised manuscript now includes the details of the precise smoothing method used in Figure 1. The p-values attached to specific tests (t, r, RR tests) are calculated usual way and such calculations normally do not form part of the manuscripts.

**Re: Point 3 -- "The authors confuse "probability" with "frequency"... and the claim that our data includes "a fixed census of data, no random sample, no experiment"**

We do not confuse probability with frequency and we do not have "a fixed census of data" if that embodies a notion that we have evaluation of every single professor. We have a sample of course evaluation obtained by a sample of professors teaching these kinds of courses. The professors in our sample were sampled usual way for these type of studies: they applied, went through interviews, and some were selected from those who applied to teach at this institution and those who taught there at those times, and got their evaluations done, formed our sample. Clearly, we worked with a sample of professors rather than all professors even if the sample was selected by others (not by us). Equally clearly, our sample is not random. However, no other sample of participants in psychology or educational studies is random. Also, many studies rely on non-experimental designs.

**Re: Point 4 -- "This is a story, not science"**

There are several separate issues here:

1) Our study shows that in this sample of professors, those teaching math do more poorly on SET than those teaching say English. This replicates findings cited in the introduction, the findings obtained in different universities. In fact, one of them is based on hundreds of institutions being evaluated by ETS student evaluation system. Thus, we replicated the previous findings that math professors do more poorly than others. This is science just as any other science that uses non-experimental designs.

2) Question as to why math profs to more poorly on SETs than English profs is not answered directly by our data nor by any other prior data which we know of. Nor did we ever attempted to answer it. Our conclusions that math profs are far less likely to be labeled unsatisfactory, fired, etc. does not hinge in any way on the reasons for the difference in the SET ratings.

3) The crux of this article is that claims that factors such as teaching math vs. English explain only 1% of variance, and thus are "not important" as was concluded in the previous research is based on inappropriate analyses and irrelevant effect sizes. When the same data are analyzed appropriately and when appropriate effect sizes are used, the difference between math and English professors is substantial and has real, huge impact on these professors being labeled satisfactory vs. non-satisfactory. Moreover, such impact depends on the specific standard used by any given institution.

4) The issue of fairness. The reviewer argues that the differences could "be due to real differences in teaching, not to subject matter per se." Agreed, it could be, even if we personally believe this is extremely unlikely given a well-established evidence that students do not like to take quantitative course, that numerical literacy is on a steep decline, etc. etc.. Moreover, the basic principles of fairness (and various codes of testing) dictate that validity of a measure as well as validity of the measure in different context (quantitative vs. non-quantitative courses) must be established before the measure is used. If it was not done, the common standards should not be used. And this goes to the next, most critical issue:

5) The validity of SETs. The reviewer then goes on and says that it is impossible "to measure teaching effectiveness" and so we cannot figure out if these two groups are engaged in equally good teaching. Well, if we cannot figure it out, and especially if we have no evidence that we can actually measure the quality of the teaching, perhaps the ONLY fair solution is to evaluate a person teaching a particular

subject against those person teaching that same subject. That way we know where they stand relative to those who try to do the same job. And corollary is that to evaluate math teachers against English teachers is unfair UNLESS we can show that math profs are really truly engaging in "bad teaching". The revised manuscript now includes more extensive discussion of these issues.

**REVIEWER 2**

**Re: Points 2 and 6: Different ways to analyze the data.**
We do not have access to data that would allow us to do meaningful analyses using multi-level modeling.

**Point 5: There could be differences in a number of students, class sizes, type of course, that could account partially for the differences in the ratings among disciplines.**
Indeed this is a possibility. However, even if this was the case, the fact remains that many universities compare one professor's SET to averages for the department and/or university, regardless of class sizes, response rates, etc..

---

## Round 0.3 · accepted · Accept

· Academic Editor

Accept

Thanks for contributing this provocative article. Congratulations. Peter

---

## Author Rebuttal · Round 0.3

# Letter

Dear Peter:

Thank you for yours and the reviewers' comments. We have now revised the manuscript following the feedback. In particular:

- We corrected grammatical and punctuation errors and had the manuscript proofread by a third person. We also rechecked the values and corrected a few mistyped values.
- We added confidence intervals on effect sizes including confidence intervals in Table 2. In Table 2, we added a column for Math vs. Non-math ratios.
- We added in the discussion a half a paragraph discussing implications of using university vs. departmental norms (in response to Reviewer 1).
- Finally, we added color to Figure 2 and 3 to better distinguish the two overlapping distribution.

The changes are marked up in blue font.

Cheers,
Bob